# Surface Behaviours of Humpback Whale *Megaptera novaeangliae* at Nosy Be (Madagascar)

**DOI:** 10.3390/biology13120996

**Published:** 2024-11-29

**Authors:** Ylenia Fabietti, Chiara Spadaro, Agnese Tigani, Gianni Giglio, Gianpiero Barbuto, Viviana Romano, Giorgio Fedele, Francesco Luigi Leonetti, Emanuele Venanzi, Carlotta Barba, Emilio Sperone

**Affiliations:** 1Department of Biology, Ecology and Earth Sciences, University of Calabria, 87036 Rende, Italy; ylenia.fabietti@unipa.it (Y.F.); chiara.spadaro93@gmail.com (C.S.); agnesetigani@gmail.com (A.T.); gianni.giglio@unical.it (G.G.); viviana.romano@unical.it (V.R.); giorgio.fedele@unical.it (G.F.); francescoluigi.leonetti@unical.it (F.L.L.); 2Department of Earth and Marine Sciences (DiSTEM), University of Palermo, Via Archirafi 22, 90123 Palermo, Italy; 3NBFC, National Biodiversity Future Center, Piazza Marina 61, 90133 Palermo, Italy; 4International Relations Division, University of Calabria, 87036 Rende, Italy; gianpiero.barbuto@unical.it; 5Manta Diving Nosy Be, Ambondrona 207, Madagascar; lelenosy@gmail.com (E.V.); carlotbarba@gmail.com (C.B.)

**Keywords:** behaviour, breeding area, humpback whale, Madagascar, whale watching

## Abstract

The surface behaviours of humpback whales were studied in the presence of a whale-watching vessel at Nosy Be (Madagascar) during whale-watching activities, in order to characterise the ethogram of these animals. On the basis of our observations, Nosy Be could be considered mainly as a wintering and weaning ground for the calves.

## 1. Introduction

Humpback whales *Megaptera novaeangliae* (Borowski, 1781) are baleen whales known for their complex songs and for the spectacular aerial nature of their surface behaviours [1]. Humpbacks are a highly migratory species and can be found in all of the world’s oceans. Their migrations are considered to be the longest known among mammals [1,2]. Indeed, they are able to travel more than 5000 miles (one way), mainly during the winter, from their feeding grounds (located at middle or high latitudes) to low-latitude zones where they breed, give birth, and wean calves [1,3,4].

Because of their surface behaviours, humpback whales are a target species for whale watching [5,6,7,8,9]. Surface behaviours occur mainly during their breeding season, making breeding areas particularly attractive to tourists (see, e.g., [10,11,12]). Whale watching industries are rapidly growing in developing countries such as Cambodia, Madagascar, Laos, Nicaragua, and Panama [10]. Nosy Be is one of these known breeding areas in the Indian Ocean (Figure 1) for the south-west population of humpback whales [13,14,15] and one of the most popular whale-watching sites for Madagascar. From July to October, this population overwinters in tropical areas after having fed in Antarctica during the summer months (November–March; [16,17]). This population is known as “C stock” by the International Whaling Commission (https://iwc.int/home, accessed on 1 October 2024), identifying the individuals that stay in Malagasy coastal waters during the breeding season as “C3 stock”. Recent satellite telemetry data have revealed that this area represents one of the main sites for a late breeding period during the winter season [15], making it of particular importance for humpback whale conservation.

Several behavioural studies have been conducted in different humpback whale breeding and wintering areas, including in South America, such as off the coasts of Ecuador (e.g., [18,19,20,21]), north-eastern Brazil [22], and Perù [5]; in Australia (e.g., [23,24]); and in Hawaiian waters (e.g., [25,26]). We focussed our studies around Nosy Be, as little is known about the surface behaviours of humpback whales in these waters. Observation of the behaviour of humpback whales during their breeding season is key to understanding the social interactions involved in reproductive behaviour, providing valuable knowledge for the implementation of conservation policies. The main aim of this study was to describe the surface behaviours of humpback whales at Nosy Be in the presence of a whale-watching boat.

## 2. Materials and Methods

### 2.1. Study Area

Individual observations were conducted in the tropical waters around Nosy Be archipelago (13°20′13″ S 48°15′37″ E), a volcanic group of islands located off the north-west coast of Madagascar (Figure 1). The island of Nosy Be is about 22.5 km long and 15 km wide with an area of 312 square km. Water depths on the continental shelf around Nosy Be are generally shallow, rarely exceeding 40 m [27,28,29]. The archipelago is a significant hotspot for marine megafauna species, including whale sharks, cetaceans, and sea turtles, as well as for coral biodiversity [27].

### 2.2. Data Collection

Data were collected from 27 July to 13 October 2018. This period coincides with the breeding season of the Antarctic population of humpback whales that arrives in Malagasy waters during the austral winter [1,3]. Observations were performed during whale-watching activities conducted by the local ecotourism operator “Manta Diving Club”, which usually departed in the morning and lasted 3 to 4 h, from ca. 900 to 1300 h. The ecotourism operator provided the facilities that allowed for data collection, including the three vessels Vandamme-designed fast boats that follow ISO standards MJM Category A “Ocean” and CE certification: “Kali” (10.5 m long, with Suzuki 250 Hp 4 stroke engine), “Samonta” (8 m long, with Suzuki 175 Hp 4 stroke engine), and “Soareziky” (7 m long, with Suzuki 150 Hp 4 stroke engine).

According to the sustainable policy of the “Manta Diving Club”, the activities took place following a code of good conduct (“Charte pour l’observation des mammifères marins”) drawn up by the local association CétaMada (https://www.cetamada.org/, accessed on 1 October 2024). The organisation’s role is to supervise the marine mammal’s observation activity to minimise the risk of impact on cetacean populations in Madagascar. Once a humpback whale was sighted, the boat headed towards it and stopped at a distance of about 150 m. When approaching the humpback whales, the approach varied depending on the current conditions and the dynamics of the aggregation. In most cases, the approach was from an oblique angle to the side in order not to surprise the animals and to have a wide observation view. Immediately, the type of aggregation was noted. Data were collected continuously, and each time the aggregation was changing behaviour, it was reported. The ethological sightings took place on board the boat [10,11,12]. They started about 2 min after the boat slowed down and stopped. The sightings ended when the whales were too far away to monitor their behaviour, and they were not chased to avoid disturbing them. The sheets were completed and eventually amended, at a later time, with the behavioural information observed through the analysis of photographs and videos captured using the personal devices of staff and customers who were present onboard during the sighting. Each behavioral unit exhibited within each aggregation was considered. If multiple individuals from the same aggregation exhibited the same behavioral unit at the same time, it was considered only once in the processing of the transition matrix and in subsequent analyses.

Four different aggregation types were observed during the field activity, as described in Table 1, according to [10,11,12].

Regarding the behavioural data collection, we followed the ethogram reported in Table 2.

### 2.3. Data Analysis

A Chi-square (χ2) test was conducted to compare the observed frequencies versus expected frequencies of the behavioural units, observed in relation to the identified groups. Four significance levels were considered: no significance (NS, *p* > 0.01), significant (*, 0.01 ≤ *p* ≤ 0.001), very significant (**, 0.001 ≤ *p* < 0.0001), and extremely significant (***, *p* ≤ 0.0001). A transition matrix was created to define the consequentiality of the different behavioural units, as well as to establish the complexity of decision-making between behaviours, which was used according to [35,36]. Statistical analyses were performed with the Instat 3.0 software, whereas the matrix was built using the EthoLog 2.2 software to aid in the transcription and timing of behavioural observation sessions from audio/video tape recordings or real-time registrations.

## 3. Results

During the three months of observations (about 400 h of data collection), 75 boat surveys were performed in total, where humpback whale encounters accounted for 68 of these trips. The most common aggregation type was G (*n* = 23; 33.82%), immediately followed by MC (*n* = 21; 30.88%), then by S (*n* = 19; 27.94%), and MCE (*n* = 5; 7.35%).

### 3.1. Behavioural Units Observed

The aggregations exhibited the behavioural units summarised in the ethogram reported in Table 2 (refer to Figure 2 for more details).

The total frequencies are reported in Table 3.

Mother–calf pairs performed all the behavioural units, except for Tail Slap (TS). They performed 80%, 78%, and 57% of total Spy-Hopping (SH), Head Slap (HS), and Logging (LOG), respectively. Tail Throw (TT) was performed 17% of the total observations of this behaviour, while Peck Slap (PS) only 2%.

Mother–calf and Escorts exhibited only four behavioural modules. They performed 48% of the total observed Breaching, while only 7% of Tail Throw. Head Slap and Peck Slap were performed with a similar percentage on the total, and Spouting with a frequency of 13%. In all Mother–calf and Escort aggregations observed at Nosy Be, Peck Slap was performed by the potential mother.

Regarding Groups, they performed Peck Slap (70%) and Spouting (48%) with high percentages on the total observed for each behaviour. Tail Slap, Breaching, and Spy-Hopping exhibited similar frequencies (10–14% on the total), while Tail Throw accounted for 28% of the total observations. Conversely, Singles performed with high frequencies Tail Slap, followed by Tail Throw and Logging. The other behavioural units showed a frequency ≤10% on the total for each behaviour.

Regarding the behavioural units, as reported in Table 3, Spouting was significantly exhibited more in Groups (χ2 = 267.47; df = 3; *p* < 0.0001) than in Mother–calf pairs, Mother–calf and Escorts, and Singles.

The aggregations that exhibited the highest Breaching frequency were Mother–calf pairs, Groups, and Mother–calf and Escorts. Singles, to the contrary, showed the lowest observation values (χ2 = 23.908; df = 3; *p* < 0.0001). HS was observed in only two aggregations: Mother–calf and Mother–calf and Escorts (χ2 = 14.309; df = 3; *p* = 0.002).

Tail Throw was observed more frequently in Singles and Groups; in the other aggregations, it was performed less so (χ2 = 9.143; df = 3; *p* = 0.02). Tail Slap was observed in only two aggregations—Singles and Groups—at relatively low values, with no significant relationship (χ2 = 6.295; df = 3; *p* = 0.09). Peck Slap was observed in all categories, with considerably higher performances and frequencies in Groups, followed by Mother–calf and Escorts. Instead, the lowest values were observed in Singles and Mother–calf pairs (χ2 = 75.222; df = 3; *p* < 0.0001). Spy-Hopping was observed in all aggregations except in Mother–calf and Escorts; its highest frequency was observed in Mother–calf pairs (χ2 = 10.445; df = 3; *p* = 0.01). Logging was performed only by Mother–calf pairs and Singles, with the highest and significant frequency in Mother–calf pairs (χ2 = 17.599; df = 3; *p* = 0.0005).

### 3.2. Ethogram and Transition Matrix

A transition matrix was obtained by processing all our data using the program EthoLog, and the result is shown in Figure 3. BL interacted with all described and observed behavioural modules in this study and preceded the others 166 times, reporting higher values for Peck Slap (38 times), Logging (36 times), and Breaching (33 times). These three behaviours showed the highest interactions after Spouting. Furthermore, Tail Throw also showed high interactions, especially with Spouting.

## 4. Discussion

Blow behaviour observed at Nosy Be was the most abundant and frequent. This is obvious because, like all mammals, humpback whales have lungs and need to breathe air at the surface, despite their adaptation to a completely aquatic life. Indeed, as the transition matrix shows (Figure 3), any action involving the emergence of the individual from the water (or part of it) is preceded by an exhalation of air and, therefore, by BL. For these reasons, this behaviour also occurs during the normal underwater swim after a dive. However, we only had the possibility to collect the data presented here from an ecotourism whale-watching boat. Utilising another method, such as unmanned aerial vehicle (UAV) technology (with or without the classical visual approaches), could improve our knowledge about the significance of the social behaviour of humpback whales for long-term studies [37].

Peck Slap (PS) exhibited the highest frequency when the correlations between behaviours and aggregations were analysed. PS behaviour is usually performed either by males as a threat demonstration against competitors or by a female to deter mating attempts [34,38]. Additionally, other studies have suggested that adult females can also exhibit PS in the presence of competitive males, probably to attract their attention and demonstrate their availability for mating [26]. Moreover, it is plausible that another function of this behaviour could be to mediate the splitting of a group in breeding areas [26]. This suggested the use of PS in close-range communication between or within a group, especially in high wind speed conditions [23]. Groups with more calves often display more PS when an escort is present, as opposed to associations devoid of escorts [38]. However, in large groups with only one calf, these behavioural events are less frequent, probably due to the calf’s presence inhibiting the PS [38]. It is worth noting, however, that the current data do not seem to support this hypothesis. Our results are similar to the ones reported in [26]. This behaviour was observed fewer times in Mother–calf pairs and only by the calf, again suggesting a form of playing activity [26]. In these pairs, mothers could also display PS to stimulate their calves’ surface activity to induce muscle development and coordination, as well as strengthening Mother–calf social bonds, as proposed in [26]. Most studies on Peck Slap have reported that its principal function is to attract the attention of one or more humpback whales, enabling the start of different social interactions [23,26,38,39]. Our data exhibited only one occurrence of PS in a Mother–calf and Escort aggregation. It is yet to be confirmed whether PS exhibits sexual dimorphism.

Spy-hopping (SH) was another behaviour observed quite frequently in Mother–calf pairs. A correlation between SH frequency in the presence of whale-watching activity has been described [7,33,40], confirming the curious nature of humpback whales. This behaviour in the mother may indicate a state of alert towards her calf; for example, SH could be used to locate possible danger such as the approach of a boat. A similar function has also been proposed for Breaching, but the slow emergence of the head upwards during SH would seem to provide the whale with a better field of view above the water line as it would leave the eyes clear from the water, compared to when it breaches. Even the slow rotation of the whale’s body outside the sea supports the hypothesis that they would look around [32]. This hypothesis could follow our data, in which the predominant frequency of SH observed within the Mother–calf aggregation suggests their curiosity towards the whale-watching boat. To the contrary, no SH was observed for Mother–calf and Escort aggregations, in relation to other, more active behaviours (e.g., BR, PS, HS, and TT), probably as all of the individuals within these aggregations were more likely to interact with each other than to consider the presence of human activities.

Logging (LOG) was performed especially by Mother–calf pairs. Some cetacean species can sleep a so-called unihemispheric slow wave sleep (USWS), whereby the animal sleeps with one brain hemisphere at a time [41]. Unlike terrestrial mammals, cetaceans are voluntary breathers. This was also seen in our whales which, at regular intervals, exhibited Spouting during LOG. While this unihemispheric sleep behaviour has been confirmed in odontocetes, it is yet to be fully understood in mysticetes [42]. A recent study identified some features in minke whales related to the control and regulation of sleep similar to those of h porpoises [43,44]. In the present study, LOG was also observed in Singles. The fact that it was not observed in aggregations with more than two individuals, for example, in Group and in Mother–calf and Escort aggregations, is intriguing. Further studies are necessary to fully comprehend sleep in mysticetes. Is it possible that these larger associations are actually less stable in time, and might occur for specific purposes (e.g., mating). It is plausible that individuals within larger groups remain more alert and avoid rest, or the reason may be found instead in the social role of a humpback whale group. It is also possible that individuals in LOG belonging to Group and Mother–calf and Escort aggregations were not observed during the present research.

Little is known about the meaning of the Head Slap (HS) behaviour: it could be attributed to a motivational change in breeding areas [38], or it could be a part of an aggressive behaviour [26]. The superficial turbulence created when the whale surfaces could be used to confuse or obscure the view of another individual [25]. In the past, some humpback whales were observed to inhale air during HS; thus, expanding the grooves of the throat to increase the body size and produce an intimidating profile [45,46]. However, this behaviour was not seen in individuals within Groups or Singles in the present study. Studies on the meaning of HS behaviour are very few: if it is to be considered as an aggressive behaviour, the absence of records in our observations could follow our hypothesis that Nosy Be could be considered a wintering area, due to the absence of competitive groups.

Tail Throw (TT) is considered an aggressive behaviour, which males exhibit as a sign of menace against other males or to attract a female, while females could use it to discourage a male’s approach [33,38]. At Nosy Be, TT events were observed several times in Groups. However, the highest frequencies occurred particularly in Singles. It is a behaviour often associated with solitary humpback whales that attempt to attract the attention of other individuals within a nearby group [33]. Other hypothesis suggests that this behaviour could represent a way to clean tail skin of parasites, in a similar way to one of the meanings hypothesised for Breaching [40,47,48]. As with Tail Throw (TT), Tail Slap (TS) may also be used to attract other individuals, especially when performed by Singles. It could be a form of non-vocal acoustic communication in humpback whales, as the sound that is generated can be localised by other individuals hundreds of metres away [39,46]. It has recently been proposed that it could play a role in close-range communication, and could express the intention to join or separate from a group [23], as previously reported for some humpback whales in Hawaiian reproduction grounds [18,34]. In Group aggregations at Nosy Be, however, TS was observed only once. A higher incidence of TS was recorded in [23] for groups where there were multiple Mother–calf pairs; thus, it could also be considered as a form of play among calves, as well as a form of social play among adults. However, Groups with multiple Mother–calf pairs were not observed in the aggregations of Nosy Be. The functions of Tail Slap vary, depending on the context and the composition of the group. Some individuals may also use it to hit each other during competitive events for access to a female [25,34]. It has been suggested that humpback whales can react with Tail Slap if bothered [49]. In the Gulf of Maine, the southern population of humpback whales slap their tails several times during feeding at the surface, probably to scare fishes and gather them in “bubble-nets” [50,51]; therefore, a foraging function for this behaviour could be also assumed.

In this study, Mother–calf and Escort and Mother–calf pair were the aggregations in which Breaching (BR) was mainly observed. These results reflect the idea that the most spectacular breaches are generally exhibited as a play function [30,31]. In [31,41,52,53], it was hypothesised that BR can be performed to signal their presence to other groups and, more likely, the sound generated by the powerful slap of the body on the surface serves to communicate between those nearby, especially when the wind speed increases and covers normal underwater vocalisations. A recent study has suggested it can be used for the communication between groups or to communicate in different acoustic conditions [23]. Breaching was observed less frequently in Singles and Groups. It was also speculated that the functions of this behaviour may be related to aggressive demonstrations among male competitors, or even as a kind of courtship. The aggressive interpretation was, however, excluded, as it seemed that the males’ attacks were more efficient through slapping of the flukes [52]. However, with BR, the males may simply display their strength to other competitors as a warning, and a female could choose according to their power during a sequence of jumps [52].

## 5. Conclusions

Studies on humpback whale behaviours are still limited, especially in breeding areas such as Nosy Be. In this study, it was not possible to identify the genders of individuals, except in Mother–calf cases: however, as Nosy Be is a breeding area, it may be plausible that there was at least one receptive female in the most active groups. Considering the difficulty in gender identification, the reproductive social role of the behaviours is still to be understood and should be thoroughly investigated. The biological function of each behaviour may vary, based on the social context and the composition of the group. However, a common denominator seems to be the communication of different motivational states of individuals (e.g., competition, to obtain attention, to play, to express an interest in joining or moving away from a group, and so on). In Groups of more than two individuals, little or no social nor aggressive behaviours were observed, probably due to a lack of needing to attract the attention of other individuals. This suggests that, during the breeding season, Nosy Be could represent a wintering and weaning ground for calves. This hypothesis is reinforced by the shallow waters around the archipelago. As previously suggested [25,36], competition would probably be more efficient when exhibiting Tail Slap, due to the slap power generated with flukes; however, at Nosy Be, this behaviour was observed less frequently than the others. We collected data from an ecotourism whale-watching boat in this study. Utilising another method, such as unmanned aerial vehicle (UAV) technology (with or without the classical visual approaches), could improve our knowledge about the significance of the social behaviour of humpback whales for long-term studies [37], as performed also for other coastal and epipelagic species [54,55].

Improving knowledge on the behaviours of humpback whales, especially in breeding areas and during whale-watching activities, is important to better understand the ecology and the life traits of these animals, as well as to identify the interactions occurring during the presence of boats. Only in this way will it be possible to arrange conservation projects to mitigate threats and protect the species.

## Figures and Tables

**Figure 1 biology-13-00996-f001:**
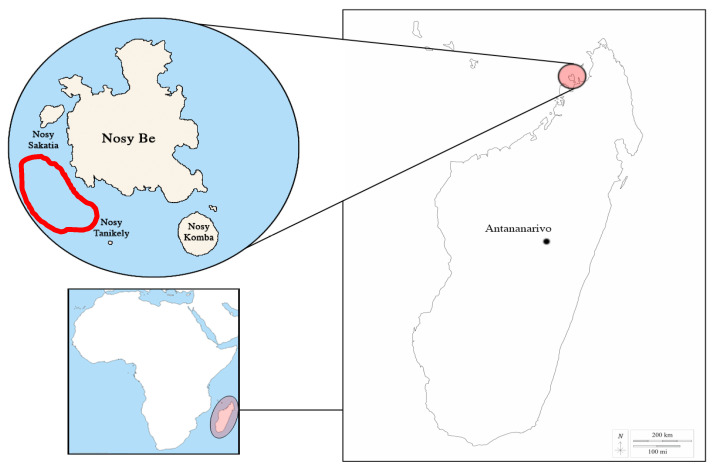
Localisation of the study area, Nosy Be (Madagascar). The red line encloses the humpback whale behaviour observation area during ecotourism activities. Coordinates of the centre of the red area: 13.362726 E, 48.159456 S.

**Figure 2 biology-13-00996-f002:**
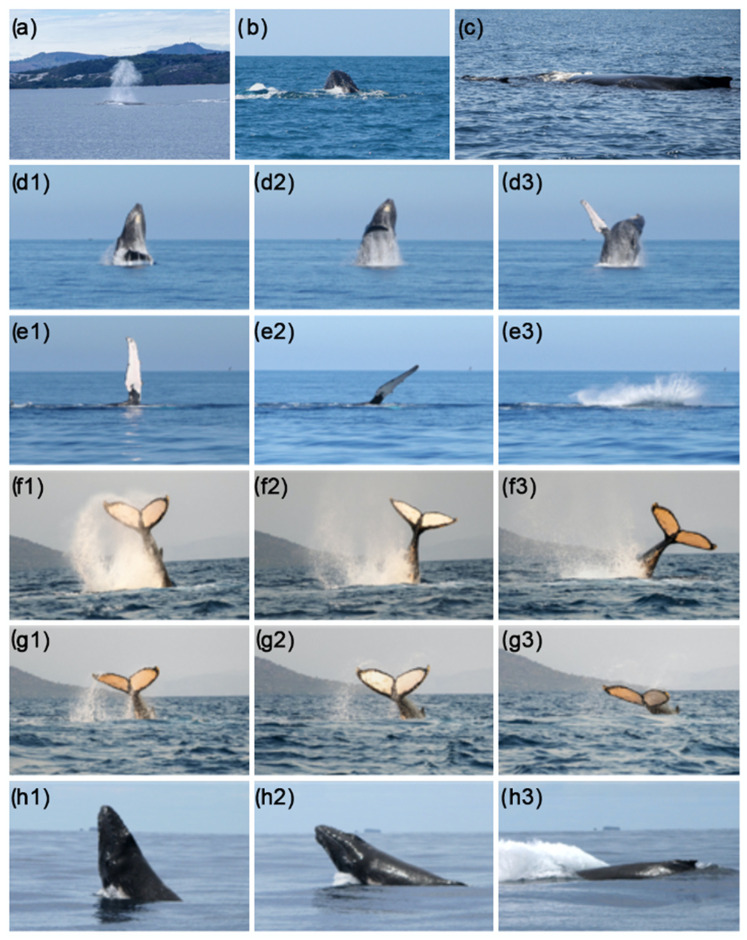
Some humpback whale surface behaviours: Spouting (BL, (**a**)), Spy-Hopping (SH, (**b**)), Logging (LOG, (**c**)), Breaching (BR, (**d1**–**d3**)), Peck Slap (PS, (**e1**–**e3**)), Tail Throw (TT, (**f1**–**f3**)), Tail Slap (TS, (**g1**–**g3**)), and Head Slap (HS, (**h1**–**h3**)).

**Figure 3 biology-13-00996-f003:**
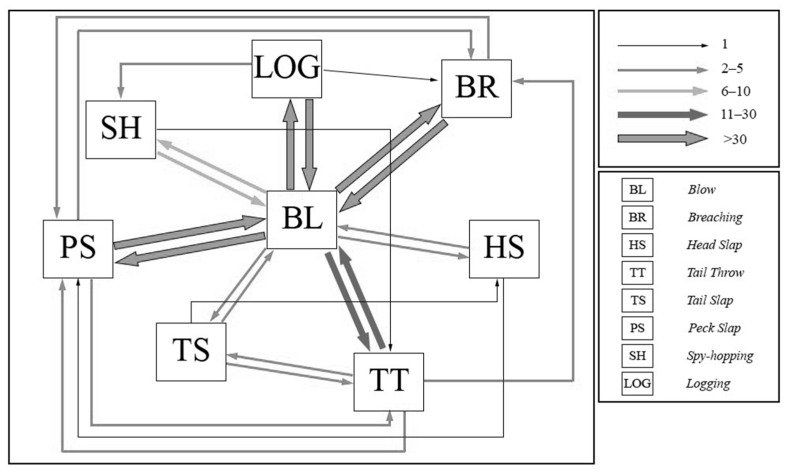
Transition matrix of the behaviours exhibited by the humpback whales observed at Nosy Be (Madagascar). Legend: BL = Spouting (Blow), SH = Spy-Hopping, LOG = Logging, BR = Breaching, PS = Peck Slap, TT = Tail Throw, TS = Tail Slap, HS = Head Slap.

**Table 1 biology-13-00996-t001:** Aggregation description observed during data collection.

Categories	Description
Singles (S)	Individual observed alone, without another visible individual within 500 m.
Mother–calf (MC)	A mother–calf pair, without another visible individual within 500 m.
Mother–calf and Escorts (MCE)	A mother–calf pair, with other individual(s) within 500 m.
Groups (G)	Two or more adults observed together or within 500 m.

**Table 2 biology-13-00996-t002:** Description of the behavioural units known, to date, in the scientific literature on the humpback whale.

Behaviour	Description
Spouting (BL)	Whales exhale air from the blowholes on top of their heads at great pressure, causing moisture in their breath to condense and create a cloud [30].
Breaching (BR)	A jump in which the animal leaves the water with at least 40% or with its entire body and lands on its back or on its side [31,32].
Head Slap (HS)	The whale raises its head out of the water, exposing the ventral furrows and slaps down violently against the surface of the water with its chin [33,34].
Logging (LOG)	The whale’s body floats on the surface in a horizontal position without showing any reduced displacement or movement [33].
Peck Slap (PS)	The animal rolls to one side or onto its back and raises one or both pectoral fins before slapping the surface with the ventral or dorsal side of the fins [33,34].
Spy-Hopping (SH)	The humpback whale raises its head vertically outside the sea surface, keeping its eyes out of the water; sometimes this behaviour is attended by a slow rotation of the body, almost as if the animal wanted to look around [31,32].
Tail Throw (TT)	Also called the “Peduncle Throw”, as the whale throws the rear part of its torso (that includes the caudal peduncle and tail) to one side above the surface of the water, creating a large splash [32].
Tail Slap (TS)	Consists of forcefully slapping the tail onto the surface of the water, either ventrally or dorsally. Normally, the whale is upside down, in a vertical position, below the surface of the water [32,33].

**Table 3 biology-13-00996-t003:** Frequencies and significance of each behavioural unit performed by humpbacks (** very significant, *** extremely significant). Legend: BL = Spouting, SH = Spy-Hopping, LOG = Logging, BR = Breaching, PS = Peck Slap, TT = Tail Throw, TS = Tail Slap, HS = Head Slap, MC = Mother–calf pairs, MCE = by Mother–calf and Escorts, S = Singles, G = Groups.

	BL	BR	HS	TT	TS	PS	SH	LOG
MC	28%	31% ***	78% **	17%	-	2%	80%	57% **
MCE	13%	48% ***	22%	7%	-	24%	-	-
S	11%	7%	-	48%	86%	4%	10%	-
G	48% ***	14%	-	28%	14%	70% ***	10%	43%
Total%	100%	100%	100%	100%	100%	100%	100%	100%

## Data Availability

Data are available on request due to restrictions, e.g., privacy or ethical.

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
