# Peer review of "Surface Behaviours of Humpback Whale Megaptera novaeangliae at Nosy Be (Madagascar)"

_biology, 2024, doi:10.3390/biology13120996_

Round 1
Reviewer 1 Report
Comments and Suggestions for Authors
The topic of the paper is interesting and it describes a population of whales not so studied. However, the aim is not clear, as there are no results about the impact of the whale watching activity. I would change the aim and the title or I would add a section of methodology and results where it is taken into account the impact of the whale watching activity on the whales (numbers of boats, distance boat-whale etc.). In my opinion the methodology is not very clear, I would describe the ethogram and the behavioural events in the methodology. Then I would compare the behavioural events for each type of aggregation (4 chi-square test, 1 for each type of aggregation). Discussion is mostly a review of the behavioural events, I would focus more on your results.

English could be improved, especially in the introduction.
Author Response
Comments 1: The topic of the paper is interesting and it describes a population of whales not so studied. However, the aim is not clear, as there are no results about the impact of the whale watching activity. I would change the aim and the title or I would add a section of methodology and results where it is taken into account the impact of the whale watching activity on the whales (numbers of boats, distance boat-whale etc.).
Response 1: We thank reviewer for this suggestion. According to other reviewers we changed aim and title since we had no enought data on whale watching activities.
Comments 2: In my opinion the methodology is not very clear, I would describe the ethogram and the behavioural events in the methodology. Then I would compare the behavioural events for each type of aggregation (4 chi-square test, 1 for each type of aggregation).
Response 2: we clarified methodology and discussed ethogram and behavioural events in the methodology. We also adjusted chi square in the text since maybe we did not wrote that part in the correct form.
Comments 3: Discussion is mostly a review of the behavioural events, I would focus more on your results
Response 3: we improved this part.
WE ALSO MADE ALL CHANGES REPORTED IN THE PDF FILE FROM THE REVIEWER
Reviewer 2 Report
Comments and Suggestions for Authors
Hi authors. First of all, congrats on the study and a such beautiful study area. I agree WW likely has no impact on the whales. However, you can't assert this based on your current analyses. If you want to make this claim, the analyses need to be refined to provide a more consistent answer. For instance, the most common surface behavior observed was spouting (blow). Humpback whales may spout less to remain "hidden", or they may spout more if they are stressed. Instead of counting spouts, you could have analyzed the intervals between spouts to see if respiratory intervals had changed. Additionally, you could have examined behavior before and during the boat's approach to determine whether behavior frequencies changed. Finally, I recommend adjusting the focus of the manuscript (make it clear), either by presenting it as only an analysis of their surface behavior (without mentioning the influence of whale watching) or by revising the analyses if you aim to evaluate a potential impact of whale watching on the whales.

It needs to be revised, especially with regard to technical terms.
Author Response
Comments 1: Hi authors. First of all, congrats on the study and a such beautiful study area. I agree WW likely has no impact on the whales. However, you can't assert this based on your current analyses. If you want to make this claim, the analyses need to be refined to provide a more consistent answer. For instance, the most common surface behavior observed was spouting (blow). Humpback whales may spout less to remain "hidden", or they may spout more if they are stressed. Instead of counting spouts, you could have analyzed the intervals between spouts to see if respiratory intervals had changed. Additionally, you could have examined behavior before and during the boat's approach to determine whether behavior frequencies changed. Finally, I recommend adjusting the focus of the manuscript (make it clear), either by presenting it as only an analysis of their surface behavior (without mentioning the influence of whale watching) or by revising the analyses if you aim to evaluate a potential impact of whale watching on the whales.
RESPONSE 1: We really thank the Reviewer for his invaluable advice, which contributed to significantly improve our paper. We have scrupulously followed all his suggestions included in the pdf he had attached. Regarding the purpose of the paper: since we had not actually collected data in such a way as to be able to speculate on the effects of the presence of boats on the behavior of humpback whales, we limited ourselves to providing an analysis of their surface behavior
Reviewer 3 Report
Comments and Suggestions for Authors
Overall statement:
The aim of this manuscript is to explore the surface behaviour of humpback whales at Nosy Be in the presence of a whale-watching boat and to assess the eventual effects of ecotourism. They found that among several measured and quantified behaviours, Blow (BL) was the most abundant and frequent.It is a relevant study and the authors touch upon an important topic of research to assess. There are increasing anthropogenic activities in marine and ocean habitats, especially in coastal areas. This paper is of interest to a broad range of behavioural ecologists and conservation biologists to increase our understanding of the potential impacts of human activities on aquatic animals here, marine mammals. I would suggest this paper be considered for publication in the journal. However, there are several structural and content-wise issues and suggestions that need to be implemented in the manuscript.
There are major and minor points in the manuscript which need clarification, refinement and rewrites and/or additional information and suggestions about what could be done to improve the manuscript.
==================================
Title:
The title is informative and relevant. Although, it is a bit too long, and I would suggest leaving" , Borowski, 1781"
================================
Abstract:
It is clear what the study found and how they did it; both the results and methodological approaches are explained.
Line:27: "The 26
The aim of the study in the abstract is not clear yet. " The surface behaviour of humpback whales was studied in the presence of whale-watching activities and vessels at Nosy Be (Madagascar) during whale-watching activities". But it needs to be clear for what reason this study has been done? For example, to explore the effects of anthropogenic activities, whale-watching vessels on the animals. As the authors mentioned in the simple summary line 15: "in order to characterize the etogram of these animals." and I would suggest incorporating this into the abstract accordingly.
Page 1, Lines:24-25: ". Individuals exhibited all the behaviours known in scientific literature: Blow,
Breaching, Head Slap, Tail Throw, Tail Slap, Peck Slap, Spy-hopping, and Logging." Please provide more explicit results rather than just counting/numbering the behavioural observations. For instance, the percentage of the behaviour per total other behavioural activities/female:male behavioural performance. The authors need to elaborate and develop this sentence accordingly.
Line:27: "The meanings of their behaviours may vary and can be multiple" The author should clarify this sentence more appropriately. What would be their suggestions and interpretation for those behavioural variations?
Line 29: remove " etc."
Lines:31-32: "The presence of the boat and the whale watching activities had little or no influence, probably due to the rules that must be respected by ecotourist operators in Madagascar. " I would suggest restructuring this statement more vigorously and based on results-driven suggestions. How do you justify that ecotourist operators in Madagascar follow and respect the rules? Are there any published evaluations or relevant regional socio-economic studies? I would suggest implementing and elaborating these latest sentences more precisely based on your invaluable behavioural results and findings.
==========================
References:
There are sentences in several sections of the manuscript which need valid/relevant references that indicated in the comments accordingly.
There are relevant references but no updated references enough. Also please include key study reference in your methodological sections and approaches.
Line 410: "(Megaptera Novaeangliae)" Scientific names should be in italic form. Please double-check all scientific names in the text but also in the reference list and edit them accordingly.
Only 6 references out of 47references are from 2020 onward. Please search for new /relevant/key references in your research theme and based on your aims and findings accordingly.
=======
Introduction/background:
The research questions are clearly outlined. Although still the research questions are not justified given what is already known about the topic. In other worlds: It is not clear what is already known about the topic in worldwide and local scale. The authors need to restructure the introduction accordingly/develop paragraphs in this regards.
Page 2, lines: 47-50: 'As with all cetaceans, humpback whales exhibit a complex plethora of conspecific social interactions: the same behaviour can have several meanings, it can be situational and it may depend on group composition. additionally, males compete intensively for access to females [5]." It seems a disconnection between before and after paragraphs here. So please elaborate this as a new independent paragraph or remove it.
Lines:51-52: "Because of their surface behaviours, humpback whales are a target species for whale 51
watching [6]. Surface behaviours occur mainly during their breeding season, making such areas particularly attractive to tourists (e.g. [7], [8], [9])" This is not a complete paragraph though. Please develop and reconstruct these paragraphs accordingly based on the topic and in a very clear story-lined sequence.
Line 54: " Nosy Be is one of the known breeding areas in the Indian Ocean for the south-west population of humpback whales [10], [11], [12]." Please add figure 1 in the text for this paragraph as you are talking about the place to refer to the relevant figure (figure 1).
Line 73: " The observations, therefore, were limited to the duration of the whale watching activity and, according to guidelines for an eco-sustainable tourism, by the possible presence of other boats in the observation area." Please provide reference for "guidelines for an eco-sustainable tourism" that you mentioned in the sentence. Is that a valid and relevant reference?
Line 76: "The behaviours were correlated to the different type 75
of aggregation, gender, and, when possible, age, in order to better understand the motivations behind a specific behaviour. "
But there is no any indication of these results in the abstract. Please update the abstract accordingly based on your results/findings.
==================================
Methods:
The process of subject selection is clear. Although how the authors detected the correct species is not delivered/mentioned yet in the section/any identification keys or reference sources.
The study methods are valid and reliable. However, the variable and behavioural indices are defined and measured appropriately, but there are no relevant and valid references to quantify them accordingly. Please elaborate and restructure these sections accordingly.
Therefore to provide an opportunity for readers and other researchers grasp the methodology well and enough detail in order to replicate the study the authors need to revise and reconstruct the whole methodological approaches/ methods used to assess the behaviour of the animals accordingly.
Line 96: "Figure 1. Localization of the study area, Nosy Be (Madagascar)" Please indicate the exact location/locations/stations of whale watching.
Line 98: "This period coincides with the breeding season of the Antarctic population of humpback whales that arrives in Malagasy waters during the austral winter. " Please provide relevant references for this sentence.
Line 100: "Observations were performed during Whale watching activities, conducted by the “Manta Diving Club”, usually from 3 to 101
4 hours in the morning, from ca. 0900 to 1300 h." please provide more detail about the “Manta Diving Club”. It was a vessel/ship? research team? What was the dimension/ HP and brand, made in which country. It should be explicitly quantified.
Line 102: Again for " The ecotourism society provided the facilities that allowed data collection: three vessels, “Kali” (10.5 m long), “Samonta” (8 m 103 long) and “Soareziky” (7 m long), are equipped with GPS, sonar, and hydrophone. T" What was the dimension/ HP and brand, made in which country.
Line 114: "The ethological sightings took place on board the boat." Please provide any relevant references.
Line 115: " They started about 2 minutes after the boat slows down and stops. " Please provide the direction of vessels/ships towards the targeted animals.
Line 123: "Aggregation description observed during data collection"
Please provide valid and relevant references for behavioural observations and classifications.
===============================
Results:
Line 190: "Table 3. Frequencies and significance of each behavioural unit performed by humpbacks (§ not quite significant, ** very significant, *** extremely significant). " There is no "§" sign on the table 3. Please double check it and edit it accordingly.
Line 202: "Figure 4. Transition matrix of the behaviours exhibited by the humpback whales observed at Nosy Be (Madagascar)."
In the figure content, what is the meaning of different colours? It sounds like a noisy figure. Is there any possibility to make it more simple and only one colour but potentially with different depth/height?
======================================
Discussion and Conclusions:
The results are not discussed from multiple angles. They need to be considered from different points of view and placed into context without being overinterpreted. For example, there are no effects of approaching vessels/ships on animals. This might also play a role. Or even shipping motor engine hp and sound produced by those at different levels may also cause impacts on the behaviour of the studied animals. These all should be considered in the discussion section and with relevant and valid references.
Line 205: "All previously described behavioural units for humpback whales were observed at Nosy Be. " This is just a one sentence and it needs to be incorporated in the paragraph. Is there any reason to separate it in one sentence? Please rewrite it accordingly.
Lines 235-236: "Most studies about Peck Slap showed that its principal function was to attract the attention of one or more humpback whales, enabling the start of different social interactions [33], [34], [23], [35], [20]. "
This is also a very short paragraph/only one long sentence. Please reconstruct the discussion section based on paragraphs accordingly.
Lines 256-257: Again the same; " Logging (LOG) was performed especially by mothers in Mother-calf, while calves 256
swam near her or showed others aerial behaviours. LOG is a behaviour in which the mother rest at the surface in proximity of her calf [32]. "
The conclusion is too extended with several paragraphs. Please just make it more brief and concise in one or two paragraphs and only, based on your findings and observations, make your conclusions accordingly.
The conclusions section do not answer the aims of the study. Please rewrite/revise this section accordingly. Also limitation of the study is not clearly outlined. A great example that the authors already covered is" in this study, it was not possible to identify the individuals’ genders, except in Mother-calf cases: however, since Nosy Be is a breeding area it may be plausible that in the most active groups there was at least one receptive female. " Please provide more examples and identify your limitations. This can help researchers/your audience and your team to consider more detail for future research and it is a kind of progress in sceince.
In overall the study design was appropriate to answer the aims. However there are several major and minor issues/comments and suggestions which are needed to be addressed and implemented in the next version of the manuscript. This study add insights to what is already known on this topic.
The manuscript is consistent within itself, but there are major flaws with this manuscript (for instance, methodological approaches are not well-defined, and no detail is provided towards reproducibility, replicability and open science.
Comments on the Quality of English Language
Please update and double check minor English grammatical points and scientific name writing in the manuscript and update it accordingly.
Author Response
Comments 1: The title is informative and relevant. Although, it is a bit too long, and I would suggest leaving" , Borowski, 1781"
Response 1: Done.
Comments 2: Line:27: "The 26
The aim of the study in the abstract is not clear yet. " The surface behaviour of humpback whales was studied in the presence of whale-watching activities and vessels at Nosy Be (Madagascar) during whale-watching activities". But it needs to be clear for what reason this study has been done? For example, to explore the effects of anthropogenic activities, whale-watching vessels on the animals. As the authors mentioned in the simple summary line 15: "in order to characterize the etogram of these animals." and I would suggest incorporating this into the abstract accordingly.
Response 2: Done
Comments 3: Page 1, Lines:24-25: ". Individuals exhibited all the behaviours known in scientific literature: Blow, Breaching, Head Slap, Tail Throw, Tail Slap, Peck Slap, Spy-hopping, and Logging." Please provide more explicit results rather than just counting/numbering the behavioural observations. For instance, the percentage of the behaviour per total other behavioural activities/female:male behavioural performance. The authors need to elaborate and develop this sentence accordingly.
Response 3: Done
Comments 4: Line:27: "The meanings of their behaviours may vary and can be multiple" The author should clarify this sentence more appropriately. What would be their suggestions and interpretation for those behavioural variations?
Response 4: we removed the sentence and clarify.
Comments 5: Line 29: remove " etc."
Response 5: Done
Comments 6: Lines:31-32: "The presence of the boat and the whale watching activities had little or no influence, probably due to the rules that must be respected by ecotourist operators in Madagascar. " I would suggest restructuring this statement more vigorously and based on results-driven suggestions. How do you justify that ecotourist operators in Madagascar follow and respect the rules? Are there any published evaluations or relevant regional socio-economic studies? I would suggest implementing and elaborating these latest sentences more precisely based on your invaluable behavioural results and findings.
Response 6: We changed the aim of the paper, accordingly also to the other reviewers suggestions, so this part has been removed
Comments 7: There are relevant references but no updated references enough. Also please include key study reference in your methodological sections and approaches.
Response 7: Done, we added more references
Comments 8: Line 410: "(Megaptera Novaeangliae)" Scientific names should be in italic form. Please double-check all scientific names in the text but also in the reference list and edit them accordingly.
Response 8: Done
Comments 9: Only 6 references out of 47references are from 2020 onward. Please search for new /relevant/key references in your research theme and based on your aims and findings accordingly.
Response 9: Done, we added what we found
Comments 10: Page 2, lines: 47-50: 'As with all cetaceans, humpback whales exhibit a complex plethora of conspecific social interactions: the same behaviour can have several meanings, it can be situational and it may depend on group composition. additionally, males compete intensively for access to females [5]." It seems a disconnection between before and after paragraphs here. So please elaborate this as a new independent paragraph or remove it.
Response 10: We removed the paragraph.
Comments 11: Lines:51-52: "Because of their surface behaviours, humpback whales are a target species for whale 51 watching [6]. Surface behaviours occur mainly during their breeding season, making such areas particularly attractive to tourists (e.g. [7], [8], [9])" This is not a complete paragraph though. Please develop and reconstruct these paragraphs accordingly based on the topic and in a very clear story-lined sequence.
Response 11: done.
Comments 12: Line 54: " Nosy Be is one of the known breeding areas in the Indian Ocean for the south-west population of humpback whales [10], [11], [12]." Please add figure 1 in the text for this paragraph as you are talking about the place to refer to the relevant figure (figure 1).
Response 12: Done
Comments 13: Line 73: " The observations, therefore, were limited to the duration of the whale watching activity and, according to guidelines for an eco-sustainable tourism, by the possible presence of other boats in the observation area." Please provide reference for "guidelines for an eco-sustainable tourism" that you mentioned in the sentence. Is that a valid and relevant reference?
Response 13: we removed the sentence since the guidelines are reported in the materials and methods section: “the code of good conduct (“Charte pour l’observation des mammifeÌ€res marins”) drawn up by the local association CétaMada (https://www.cetamada.org/)”.
Comments 14: Line 76: "The behaviours were correlated to the different type 75
of aggregation, gender, and, when possible, age, in order to better understand the motivations behind a specific behaviour. "But there is no any indication of these results in the abstract. Please update the abstract accordingly based on your results/findings.
Response 14: done!
Comments 15: The process of subject selection is clear. Although how the authors detected the correct species is not delivered/mentioned yet in the section/any identification keys or reference sources. The study methods are valid and reliable. However, the variable and behavioural indices are defined and measured appropriately, but there are no relevant and valid references to quantify them accordingly. Please elaborate and restructure these sections accordingly. Therefore to provide an opportunity for readers and other researchers grasp the methodology well and enough detail in order to replicate the study the authors need to revise and reconstruct the whole methodological approaches/ methods used to assess the behaviour of the animals accordingly.
Response 15: Done, we rearranged and improved the section
Comments 16: Line 96: "Figure 1. Localization of the study area, Nosy Be (Madagascar)" Please indicate the exact location/locations/stations of whale watching.
Response 16: Done
Comments 17: Line 98: "This period coincides with the breeding season of the Antarctic population of humpback whales that arrives in Malagasy waters during the austral winter. " Please provide relevant references for this sentence.
Response 17: Done
Comments 18: Line 100: "Observations were performed during Whale watching activities, conducted by the “Manta Diving Club”, usually from 3 to 101
4 hours in the morning, from ca. 0900 to 1300 h." please provide more detail about the “Manta Diving Club”. It was a vessel/ship? research team? What was the dimension/ HP and brand, made in which country. It should be explicitly quantified.
Response 18: Done, MantaDiving is the ecotourism society that hosted pur researchers onborad
Comments 19: Line 102: Again for " The ecotourism society provided the facilities that allowed data collection: three vessels, “Kali” (10.5 m long), “Samonta” (8 m 103 long) and “Soareziky” (7 m long), are equipped with GPS, sonar, and hydrophone. T" What was the dimension/ HP and brand, made in which country.
Response 19:Done
Comments 20: Line 114: "The ethological sightings took place on board the boat." Please provide any relevant references.
Response 20: Done
Comments 21: Line 115: " They started about 2 minutes after the boat slows down and stops. " Please provide the direction of vessels/ships towards the targeted animals.
Response 21: Done
Comments 22: Line 123: "Aggregation description observed during data collection" Please provide valid and relevant references for behavioural observations and classifications.
Response 22: Done
Comments 23: Line 190: "Table 3. Frequencies and significance of each behavioural unit performed by humpbacks (§ not quite significant, ** very significant, *** extremely significant). " There is no "§" sign on the table 3. Please double check it and edit it accordingly.
Response 23: Done
Comments 24: Line 202: "Figure 4. Transition matrix of the behaviours exhibited by the humpback whales observed at Nosy Be (Madagascar)." In the figure content, what is the meaning of different colours? It sounds like a noisy figure. Is there any possibility to make it more simple and only one colour but potentially with different depth/height?
Response 24: Done
Comments 25: The results are not discussed from multiple angles. They need to be considered from different points of view and placed into context without being overinterpreted. For example, there are no effects of approaching vessels/ships on animals. This might also play a role. Or even shipping motor engine hp and sound produced by those at different levels may also cause impacts on the behaviour of the studied animals. These all should be considered in the discussion section and with relevant and valid references.
Response 25: We rearranged discussion also according the revisions suggested for the general aim of the paper by the reviewers
Comments 25: Line 205: "All previously described behavioural units for humpback whales were observed at Nosy Be. " This is just a one sentence and it needs to be incorporated in the paragraph. Is there any reason to separate it in one sentence? Please rewrite it accordingly.
Response 26: Done
Comments 26: Lines 235-236: "Most studies about Peck Slap showed that its principal function was to attract the attention of one or more humpback whales, enabling the start of different social interactions [33], [34], [23], [35], [20]. "This is also a very short paragraph/only one long sentence. Please reconstruct the discussion section based on paragraphs accordingly.
Response 27: Done
Comments 28: Lines 256-257: Again the same; " Logging (LOG) was performed especially by mothers in Mother-calf, while calves 256
swam near her or showed others aerial behaviours. LOG is a behaviour in which the mother rest at the surface in proximity of her calf [32]. "The conclusion is too extended with several paragraphs. Please just make it more brief and concise in one or two paragraphs and only, based on your findings and observations, make your conclusions accordingly.
Response 28: Done
Comments 29: The conclusions section do not answer the aims of the study. Please rewrite/revise this section accordingly. Also limitation of the study is not clearly outlined. A great example that the authors already covered is" in this study, it was not possible to identify the individuals’ genders, except in Mother-calf cases: however, since Nosy Be is a breeding area it may be plausible that in the most active groups there was at least one receptive female. " Please provide more examples and identify your limitations. This can help researchers/your audience and your team to consider more detail for future research and it is a kind of progress in sceince.
Response 29: We revised accordingly to the previous revisions.
Round 2
Reviewer 1 Report
Comments and Suggestions for Authors
The research paper is improved since the first review but, in my opinion, there are still some corrections to do and some parts to clarify before the publication. The two main problems that I see here are: 1) the data collection (it is not clear how you considered the behavioral units) and 2) the analyses of frequencies (I would change the type of analyses, see the comments below). In my opinion the paper can be published only if these two problems are solved.
I report here my comments:
Title: I would remove “in Presence of a Whale-Watching Vessel”. You are describing the behavioral units of humpback whales. The fact that you collected the data from a whale watching boat is not relevant for your study. You can mention in the methodology but the title could be misinterpreted.
Introduction: I would add some information about the humpback whales’ behavior and the aggregation types. In your introduction you focused on some topics that you are not considering in your study (for example the migration).
Lines 57-59: I would rephrase the sentence.
Lines 80-81: I would move the aim of your study at the end of the introduction.
Lines 92-100: I would remove this part.
Materials and methods
Lines 107-111: I would remove this part. Information about temperature and tide is not relevant for your study.
Lines 143-153: Here it is not clear how you collected the data. How did you count the behavioral units? For example, in a group of 4 whales, any behavioral unit shown by one of the 4 whales was counted? If each of the 4 whales slaps the tail one time, did you count 4? How can you compare then the number of units exhibited by a big group of whales with a single whale? Please specify in the text.
Line 152: I would be consistent in all the manuscript. You used the term “aggregation types”, so I would use that word everywhere.
Lines 163-167: as I said in the previous review, I would do a chi-square test for each type of aggregation. You will have 4 results (1 for each aggregation type) and you can just give the % of each behavioral unit.
Lines 165-166: You wrote “The frequencies were calculated by dividing the number of each behavioural unit exhibited by the type of association by the total number of behavioural units exhibited by the association”.
If I understood well, for example, if the singles were doing 100 units in total and 50 of them were spy hopping, the frequency of spy-hopping was 50%, correct? Why then, in your table 3, didn’t you show frequencies like this?
Results
Lines 179-185: It is fine, maybe I would not use decimal numbers so for example, G (n = 23; 34%).
Line 192: when you say that mother-calf were performing spy hopping what do you mean? Was the mother or the calf? Or did you just consider any of them doing spy-hopping?
Lines 193-195: the sentence is confusing because it seems that the mother-calf pairs spent 80% of their time doing SH, 78% doing HS, and 57% doing LOG. But if I understood you mean that 80% of the SH units that you recorded were made by mother and calf pairs. First, I would not mention the word “time” because you are considering short behavioral units and not behavioral categories like swimming or resting. Saying that SH occurs 80% of the time means that whales in, for example 100 hours, spend 80 hours spy hopping and it is not correct. Here you are counting the number of behavioral units. Second, the numbers you are giving are from a different calculation. You should consider how many times each unit have been exhibited by mother-calf pairs. For example, mother-calf exhibit 200 behavioral units in total: 100 SH, 50 HS and 50 LOG. The frequencies will be 50% SH, 25% HS and 25% LOG. I would do that for all the aggregation types and I would change table 3.
Table 3: As I commented in the previous review in my opinion you should change the analyses and do a different comparison. For example, in the table you are saying that 48% of the spouting units were exhibited by groups of whales (2 or more) while only the 11% by single whales. But a group of many whales will probably exhibit more units than a single whale. So how can you compare these 2 aggregation types? Why don’t you compare the units exhibited by the groups and then the units exhibited by the singles in two different tests?
Lines 238-244: Why did you do the interaction matrix not divided per aggregation types?
Discussion: The discussion needs to be rewritten according to the new analyses. In your results you don’t see what are the most common units of each aggregation type. For example, what is the most common behavioral unit exhibited by the mother-calf pairs? In your result you consider SH the most common unit with 80%. But that 80% is compared with the SH units of the other aggregation types. So, you can say that mother-calf pairs exhibit more SH than singles and groups but you don’t know if the most common behavior unit is SH. In the discussion you mention that: “ BL was the most abundant and frequent”. The interesting result here is to see if the behavioral units are different according to aggregation types. Redo the analyses showing what is the most frequent unit and then rewrite the discussion accordingly.
Lines 257-260: Here you are mentioning results that are not shown in the results section. If you consider G as a single category you cannot now in the discussion mention different types of G.
Lines 279-280: The discussion in my opinion needs to be rewritten according to your results. In the discussion you are mentioning things that have not been shown in the results section. For example, this is what you wrote in the discussion “In all Mother-–calf and Groups Escorts aggregations observed at Nosy Be, PS was performed by the potential mother and more often by the calf as a form of play”.
This is what you wrote in the results about the PS: “PS was observed in all categories, with considerably higher performances and frequencies in Groups, followed by Mother-–calf and Escorts. Instead, the lowest values were observed in Singles and Mother-–calf pairs (χ2 = 75.222; df = 3; p < 0.0001).” The two sections are not bonded together.
Lines 295-296: here again you say in the discussion: “Spy-hopping (SH) was another behaviour observed quite frequently in Mother-–calf pairs and, exhibited especially by calves”. But in the results you are not considering mother and calf separately, you just wrote that the “highest frequency was observed in Mother-–calf pairs (χ2 = 222 10.445; df = 3; p = 0.01)
Comments on the Quality of English LanguageThe text should be checked by some english native speaker. Many sentences are not very clear and need some rephrasing.
Author Response
Comments 1: Title: I would remove “in Presence of a Whale-Watching Vessel”. You are describing the behavioral units of humpback whales. The fact that you collected the data from a whale watching boat is not relevant for your study. You can mention in the methodology but the title could be misinterpreted.
Response 1: Done
Comments 2: Introduction: I would add some information about the humpback whales’ behavior and the aggregation types. In your introduction you focused on some topics that you are not considering in your study (for example the migration).
Response: We thank the reviewer for this observation. We have inserted, as suggested by the other reviewers, the description of the aggregations and behavioral modules in the MATERIALS AND METHODS section (lines 154-161). We consider the part relating to migrations important for readers who are not familiar with the life cycle of these whales, to understand the role of Nosy Be as a breeding area. However we removed lines 57-59 as they provided unnecessary information.
Comments 2: Lines 57-59: I would rephrase the sentence.
Response: we removed this part.
Comments 4: Lines 80-81: I would move the aim of your study at the end of the introduction.
Response: Done
Comments 5: Lines 92-100: I would remove this part.
Response: Done
Materials and methods
Comments 6: Lines 107-111: I would remove this part. Information about temperature and tide is not relevant for your study.
Response: Done
Comments 7: Lines 143-153: Here it is not clear how you collected the data. How did you count the behavioral units? For example, in a group of 4 whales, any behavioral unit shown by one of the 4 whales was counted? If each of the 4 whales slaps the tail one time, did you count 4? How can you compare then the number of units exhibited by a big group of whales with a single whale? Please specify in the text.
Response: Done. We added: “Each behavioral unit exhibited within each aggregation was considered. If multiple individuals from the same aggregation exhibited the same behavioral unit at the same time, it was considered only once in the processing of the transition matrix and in subsequent analyses.”
Comments 8: Line 152: I would be consistent in all the manuscript. You used the term “aggregation types”, so I would use that word everywhere.
Response: Done
Comments 9: Lines 163-167: as I said in the previous review, I would do a chi-square test for each type of aggregation. You will have 4 results (1 for each aggregation type) and you can just give the % of each behavioral unit.
Response: We thank the reviewer for this observation. We have not recalculated the chi-square values ​​on the total for each group because in that case only the BL would be significant, which would be the behavior exhibited predominantly. On the other hand, the significance with which the behaviors exhibited less often are distributed within each association would not be appreciated.
Comments 10: Lines 165-166: You wrote “The frequencies were calculated by dividing the number of each behavioural unit exhibited by the type of association by the total number of behavioural units exhibited by the association”.
If I understood well, for example, if the singles were doing 100 units in total and 50 of them were spy hopping, the frequency of spy-hopping was 50%, correct? Why then, in your table 3, didn’t you show frequencies like this?
Response: We removed the sentence, as it was a typo from a previous version
Results
Comments 11: Lines 179-185: It is fine, maybe I would not use decimal numbers so for example, G (n = 23; 34%).
Response: For better data accuracy, we would prefer to keep decimals.
Comments 12: Line 192: when you say that mother-calf were performing spy hopping what do you mean? Was the mother or the calf? Or did you just consider any of them doing spy-hopping?
Response: According to lines 143-153, We consider any of them doing spy-hopping
Comments 13: Lines 193-195: the sentence is confusing because it seems that the mother-calf pairs spent 80% of their time doing SH, 78% doing HS, and 57% doing LOG. But if I understood you mean that 80% of the SH units that you recorded were made by mother and calf pairs. First, I would not mention the word “time” because you are considering short behavioral units and not behavioral categories like swimming or resting. Saying that SH occurs 80% of the time means that whales in, for example 100 hours, spend 80 hours spy hopping and it is not correct. Here you are counting the number of behavioral units. Second, the numbers you are giving are from a different calculation. You should consider how many times each unit have been exhibited by mother-calf pairs. For example, mother-calf exhibit 200 behavioral units in total: 100 SH, 50 HS and 50 LOG. The frequencies will be 50% SH, 25% HS and 25% LOG. I would do that for all the aggregation types and I would change table 3.
Response: We thank the reviewer for this observation. In fact, we expressed ourselves badly in presenting the results. We preferred to try to be more explicit (modifying the text and adding the total to table 3 to help the reader understand the value of the percentages better). We did not recalculate the percentages on the total for each group because in that case we would only see high percentages for the BL (as is obvious) and we would not appreciate the frequencies with which the behaviors with fewer absolute values ​​are exhibited.
Comments 14: Table 3: As I commented in the previous review in my opinion you should change the analyses and do a different comparison. For example, in the table you are saying that 48% of the spouting units were exhibited by groups of whales (2 or more) while only the 11% by single whales. But a group of many whales will probably exhibit more units than a single whale. So how can you compare these 2 aggregation types? Why don’t you compare the units exhibited by the groups and then the units exhibited by the singles in two different tests?
Response: See Response 7
Comments 15: Lines 238-244: Why did you do the interaction matrix not divided per aggregation types?
Response: we preferred to build a single matrix to highlight the general functional transition between behaviors, regardless of the type of aggregation they exhibit. The too low numbers of some behavioral units do not allow us at the moment to better detail this part
Comments 16: Discussion: The discussion needs to be rewritten according to the new analyses. In your results you don’t see what are the most common units of each aggregation type. For example, what is the most common behavioral unit exhibited by the mother-calf pairs? In your result you consider SH the most common unit with 80%. But that 80% is compared with the SH units of the other aggregation types. So, you can say that mother-calf pairs exhibit more SH than singles and groups but you don’t know if the most common behavior unit is SH. In the discussion you mention that: “ BL was the most abundant and frequent”. The interesting result here is to see if the behavioral units are different according to aggregation types. Redo the analyses showing what is the most frequent unit and then rewrite the discussion accordingly.
Response: in redoing the analyses as suggested by the reviewer, it would appear that each group performs only the BL with high frequencies and the detail of the distribution of the other behavioral units would be lost. For this reason we prefer to keep the analyses as we set them, also because this was appreciated and integrated by the other two reviewers
Comments 17: Lines 257-260: Here you are mentioning results that are not shown in the results section. If you consider G as a single category you cannot now in the discussion mention different types of G.
Response. We removed the paragraph
Comments 18: Lines 279-280: The discussion in my opinion needs to be rewritten according to your results. In the discussion you are mentioning things that have not been shown in the results section. For example, this is what you wrote in the discussion “In all Mother-–calf and Groups Escorts aggregations observed at Nosy Be, PS was performed by the potential mother and more often by the calf as a form of play”.
This is what you wrote in the results about the PS: “PS was observed in all categories, with considerably higher performances and frequencies in Groups, followed by Mother-–calf and Escorts. Instead, the lowest values were observed in Singles and Mother-–calf pairs (χ2 = 75.222; df = 3; p < 0.0001).” The two sections are not bonded together.
Response: We removed all paragraphs discussing things non included in the results section
Comments 19: Lines 295-296: here again you say in the discussion: “Spy-hopping (SH) was another behaviour observed quite frequently in Mother-–calf pairs and, exhibited especially by calves”. But in the results you are not considering mother and calf separately, you just wrote that the “highest frequency was observed in Mother-–calf pairs (χ2 = 222 10.445; df = 3; p = 0.01)
Response: We removed from the discussion
Reviewer 3 Report
Comments and Suggestions for Authors
The current version is now revised and met almost all comments provided by the reviewer.
Author Response
Comments 1: The current version is now revised and met almost all comments provided by the reviewer.
Response 1: We thank the reviewer for his/her appreciation and his/her suggestions